# Contractile and mechanical properties of hamstring muscles measured by the method of tensiomyography (TMG) in professional soccer players: A systematic review, meta-analysis and meta-regression

**Daniel Fernández-Baeza** [ORCID]*⊕, **Cristina González-Millán**⊕, **Germán Díaz-Ureña** [ORCID]⊕

Department of Health, Universidad Francisco de Vitoria, Madrid, Spain

⊕ These authors contributed equally to this work.

* d.fbaeza.prof@ufv.es

## Abstract

Tensiomyography (TMG) is a non-invasive device used to assess contractile and mechanical properties during an isometric muscle contraction. The purpose of this systematic review was two-fold: 1) to know the scientific evidence of hamstring muscles TMG parameters in professional football players during the competitive season, and 2) to establish the most-frequent values for the main TMG parameters in soccer players compared with the reference values of the TMG software. PRISMA guidelines were followed, and a systematic search was performed in the PubMed, Web of Science, and Sport Discus electronic databases with no date restrictions until November 2023. The studies considered for this meta-analysis were studies investigating professional soccer players between 20- and 29-years measured during the competitive season and reported tensiomyography-derived parameters such as contraction time and/or maximal displacement, and/or delay time, of the hamstring muscles (biceps femoris and semitendinosus). A total of 139 studies were identified and 12 studies were included in the systematic review and for the meta-analysis. All studies underwent a quality assessment using the Newcastle–Ottawa scale, and the NOS score varied from 7/9 to 8/9 in all studies, suggesting a good quality of all articles. Study results were analyzed using restricted maximum-likelihood and random-effects models. The main findings of the study are that of the six parameters analysed, three variables were found to differ significantly. Furthermore, the weighted mean values founded were biceps femoris (Tc 27.88, Dm 5.2, Td 23.72) and in semitendinosus (Dm 8.72, Td 25.25). TMG can serve as a valuable device for assessing neuromuscular function in soccer players. Furthermore, shows the most-frequent values of the biceps femoris and the semitendinosus, where different values in the TMG parameters can be observed between the synergistic muscles.

provided the original author and source are credited.

**Data availability statement:** All relevant data are within the manuscript and its Supporting information files.

**Funding:** The author(s) received no specific funding for this work.

**Competing interests:** The authors have declared that no competing interests exist.

## Introduction

The assessment of muscle function is crucial for planning programs that enhance sports performance and help prevent injuries [1]. In this regard, tensiomyography (TMG) has become an established non-invasive and objective tool for measuring the mechanical and contractile characteristics of muscles over the past twenty years [2,3].

TMG generates electrical stimulation in the superficial muscles, providing parameters that inform us about muscle contraction time (Tc), muscle displacement (Dm), which indicates muscle tone, and variables related to fatigue: relaxation time (Tr) and maintenance time of contraction (Ts) [4,5]. Furthermore, TMG provides information that allows us to identify lateral asymmetries (between both legs) and functional imbalances, for example, between agonist and antagonist muscles (hamstrings/quadriceps) [6].

Regarding the contraction time, it has been observed that this parameter can be a good indicator of the prevalence or not of fast twitch fibers [7]. Shorter Tc is considered to reflect a higher rate of force production [8]. Regarding Dm, it is considered to reflect muscle belly stiffness [9] and has been shown to be altered with muscle fatigue and ageing [8,10]. Dm and Tr have also been shown to rise with increasing muscle fatigue [11]. Different training states also reflect variations in TMG, with athletes with prevalence of strength and power training having shorter Tc and smaller Dm [12–14]. Likewise, it has been observed that Dm increases when atrophy occurs in muscle architecture [15,16].

In recent years hamstring injuries in soccer have increased by 4% [17]. In a longitudinal study from 2001 - 2022, hamstring injuries constitute 24% of all injuries in elite UEFA teams [18]. In the study by Dordević et al. [19], significant differences in Tc were observed between injured and non-injured biceps femoris muscles, indicating a strong predictive capacity of the Tc parameter. It has been observed that when the Dm of the semitendinosus (ST) muscle is very high, if a training programme is applied to reduce it, hamstring injuries are reduced [20]. For more severe pathologies, such as ACL injury, an increase in Tr has been observed, indicating a lower fatigue strength [21].

Numerous studies have used TMG to assess soccer players [22–29]. However, it was not until this year that an interesting study conducted a systematic review of the literature with the aim of comprehensively synthesizing the existing studies that have reported on TMG-derived parameters of the lower limbs in soccer players [30]. Regarding the use of TMG in injury prevention, to date, there is only one experimental study in which the number of hamstring injuries was reduced after implementing an individualized training program for each player based on TMG data [20]. Another issue is that there is no consensus on what the appropriate values for a professional football player should be. The TMG software provides specific reference values for all parameters, which represent the averages of the values most frequently encountered in the assessments performed by the software developers (TMG-BMC Ltd., Ljubljana, Slovenia). Regarding these values, in the hamstring muscles, we have observed that the reference value for the ST, both for Tc and Dm, differs significantly from the values of the biceps femoris (BF). In the study we conducted, which we mentioned previously [20], we observed that bringing the ST values closer to those of the BF reduced the number of hamstring injuries. Additionally, the reference data provided by the software does not distinguish between the playing levels of football players, such as professionals and amateurs. Furthermore, it is worth noting that the financial cost of having injured players leads to economic losses and a decline in team performance, which is ultimately reflected in the standings. Therefore, the aim of this article was twofold: 1) to examine the scientific evidence on TMG parameters of hamstring muscles in professional soccer players during the competitive season, and 2) to establish the most frequent values of hamstring muscles for the main TMG parameters in soccer players in comparison with the reference values provided by the TMG software.

## Materials and methods

A systematic revision was conducted November 10th to December 22nd, 2023 to search the published scientific evidence of the contractile and mechanical properties of healthy professional soccer players during the competitive season. The reporting flow diagram of this systematic review was based on the Preferred Reporting Items for Systematic reviews and Meta-Analyses (PRISMA) guidelines [31] (Fig 1). This review has been recorded in PROSPERO (CRD42024505107).

### Search strategy

The authors independently performed a literature search in PubMed, Web of Science, and Sport Discus electronic databases. In addition, the authors conducted a thorough search of the

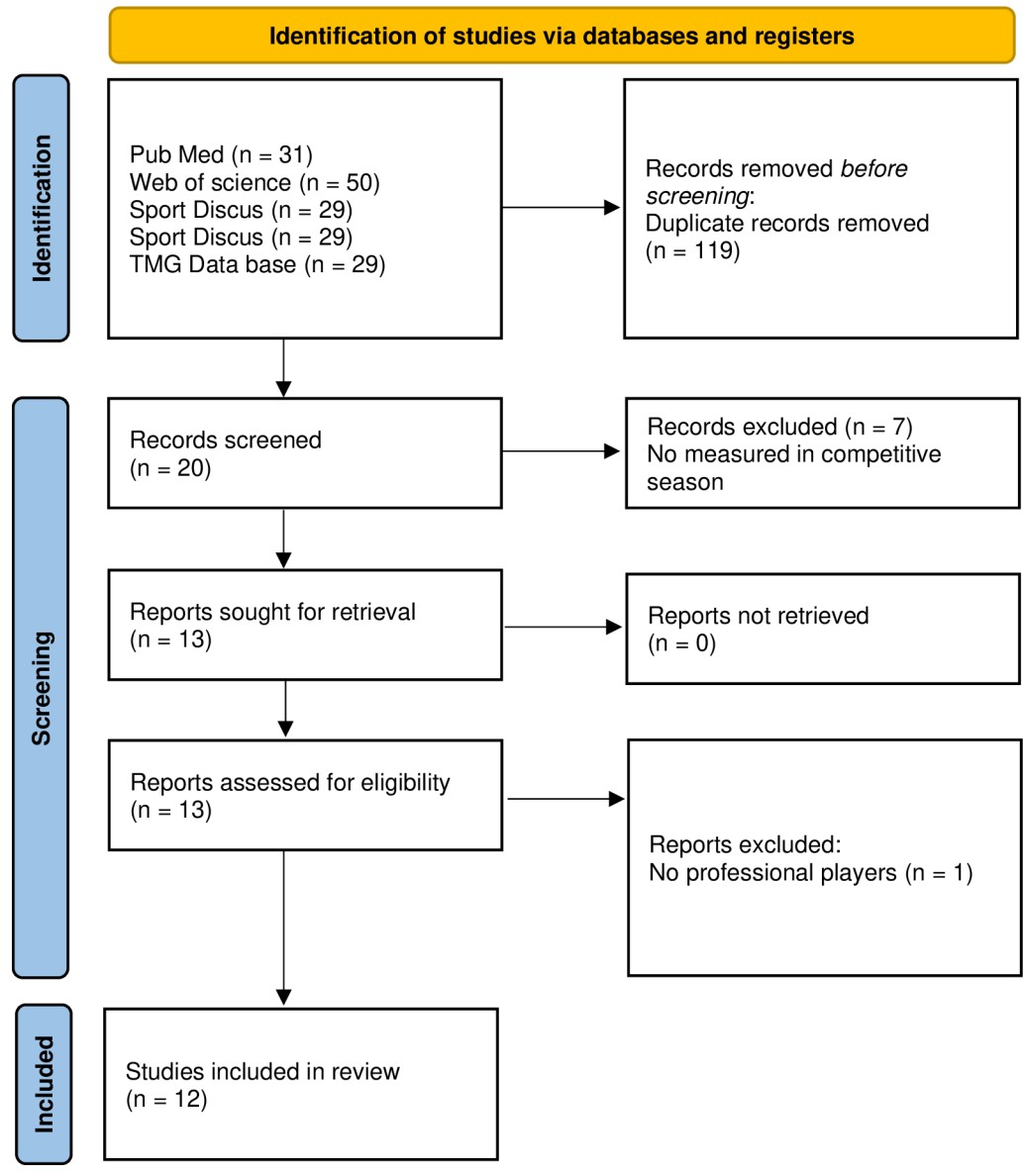

**Fig 1. PRISMA flow diagram of included studies.**

TMG-BMC website (TMG-BMC Ltd, 2022) to find additional articles. The following terms and their combinations were used as a search strategy string: (a) target population: "soccer" or "football" and "player," or "; (b) intervention: "tensiomyography," or "TMG"; (c) outcome corresponded to tensiomyography assessment. Inclusion criteria consisted of samples of professional male soccer players measured during the competitive season. The authors also consulted other experts in the field to identify any additional published studies.

## Eligibility criteria

Eligible studies were selected based on the criteria and were defined as follows healthy soccer or football players; the use of TMG; all type of studies were included in this review, except case studies and reviews. Overall, the selected studies focused on changes in TMG-derived parameters.

Inclusion criteria were male, ages between 20 to 29 years, professional soccer player, measured in competitive season. Women football players were not included due to the significant differences in physical, physiological, and biomechanical characteristics between male and female football players. Since the focus of the study is to homogenize the sample to more precisely analyze the impact of these variables on male football players, including women could introduce biases that would compromise the internal validity of the findings.

Studies that reported an inappropriate population for this review were excluded. Studies with a sample of teenagers and players with more than 29 years old were excluded-

## Study selection and data extraction

Titles and abstracts from the electronic searches were screened independently by the authors. The authors (D-F and G-D) then checked the full texts of selected articles to consider the fit with eligibility criteria. In case of any disagreement between the authors, a third reviewer (C-G) was consulted to make a final decision. The authors (D-F and G-D) separately extracted data. In the case of a disagreement, the third author cross-examined doubtful data. The following data was extracted: authors, year of publication, study population (sample size and age), and the TMG- parameters of the knee flexor muscles: biceps femoris and semitendinosus. These two muscles provide a more specific and relevant assessment of the contractile and mechanical properties that directly impact athletic performance and injury prevention. The TMG variables extracted were three: contraction time (Tc), this parameter is fundamental for analyzing how quickly a muscle can generate force, directly influencing performance. Delay time (Td) because this is crucial for understanding muscle activation speed, which is essential in sports, where reaction time is key. An optimal Td in the hamstrings ensures quick responses to changes in pace and direction, which are essential in unpredictable game situations. Muscle displacement (Dm) a key indicator of its contraction and recovery capacity, it is relevant for monitoring muscle fatigue and assessing the risk of injury. These three parameters (Td, Tc, Dm) allow for a comprehensive evaluation of muscle performance and injury prevention, aligning directly with the study's objectives in elite athletes. The relaxation time (Tr) and sustain time (Ts) variables were not extracted because, while complementary, they are not as crucial for evaluating the central aspects of muscle performance that are of greatest interest in this study. In case of missing data, the authors of the publications were contacted. Missing data were addressed through imputation methods to maintain data integrity and minimize bias cases with unresolvable gaps were excluded from specific analyses, ensuring consistency in the reported results We have used data for a dominant (kicking) leg for the analysis. The TMG software calculated the average values with the specific muscle contractile and mechanical properties based on gender, age, and sport/position, regardless of laterality and dominant

side. A weighted averaging technique was used to obtain the pooled average and standard deviation of Tc, Dm, and Td.

## Statistical analyses

The weighted mean and standard deviation were calculated using sample sizes of each study. In addition, the mean value and standard deviation of the control group were calculated from the weighted mean value and standard deviation based on the sample size of the three sub-groups from the TMG software. The standardized mean difference was used as the outcome measure, with data fitted to a random-effects model. Heterogeneity analysis involved estimating $tau^2$ using the restricted maximum-likelihood estimator. Additionally, Q-test and $I^2$ statistics are reported. To assess potential outliers or influential studies, studentized residuals and Cook's distances are applied within the model context. Funnel plot asymmetry was examined using the rank correlation test (Begg and Mazundar correlation test) and the regression test (Egger regression), utilizing the standard error of the observed outcomes as a predictor. Studies which altered homogeneity outcome were removed from the analysis of each variable. Cohen´s kappa (κ) were calculated in order to measure the inter-rater reliability for the two authors who select the studies. The normalized reaction velocity (Vrn) was calculated with the formula: 0.8/Tc [22]. To minimize the risk of overfitting, a meta-regression was performed using age as a covariate. The p-value was set at 0.05. Statistical analysis was conducted using RStudio v. 2023.09.1 + 494 and Meta package v.6.5.

## Methodological quality assessment

The quality of the selected studies was assessed according to the Newcastle–Ottawa scale (NOS), and the NOS score varied from 7 to 8 in all studies, suggesting a good quality of all articles.

## Results

Table 1 shows the results for all combinations used to carry out this review.

A flow chart of study selection is shown in Fig 1.

Due to the heterogeneity of the results, it was not possible to perform the analysis of the ST contraction time. However, all TMG values presented in all studies found in this review have higher values than the TMG Software values. Bias assessments are shown in Table 2. Neither the rank correlation nor the regression test indicated any funnel plot asymmetry on any of the variables.

**Table 1. Specific search equation results.**

|  | PubMed | Sport discus | Web of science |
|---|---|---|---|
| Soccer and TMG | 20 | 20 | 34 |
| Soccer and tensiomyography | 28 | 27 | 52 |
| Football and TMG | 8 | 12 | 16 |
| Football and Tensiomyography | 10 | 15 | 24 |
| (((Soccer)) OR (football)) AND ((TMG))) | 22 | 16 | 36 |
| (((Soccer)) OR (football)) AND ((tensiomyography))) | 31 | 16 | 55 |
| (((Soccer players)) OR (football players)) AND ((tensiomyography))) | 31 | 10 | 50 |
| (((Soccer players)) OR (football players)) AND ((TMG))) | 22 | 10 | 32 |
| (((soccer players)) OR (football players)) AND ((TMG) OR (tensiomyography))) | 31 | 29 | 50 |

Table 3 shows heterogeneity statistics of each variable. Tc BF shows a high heterogeneity. This result means that we did not include it in the meta-analysis. Other variables show low heterogeneity (I2 < 25%, H2 ≅ 1, p > 0.05) except Td ST which shows a low-moderate heterogeneity (I2 = 34.43%, H2 ≅ 1, p > 0.05).

Table 4 shows values of the random-effects model. Only Td values have no significant differences. The average outcome of most variables, except Dm BF, were positive $(\breve{\mu} - value)$ . In addition, these results were significant differences in Tc BF, Dm BF and Dm ST. The higher the positive values, the higher the values in the studies related to TMG.

Table 5 shows values of the meta regression mixed-effects model using age as covariable. No significance effects were observed.

Fig 2 and 3 show forest plot of each value included in this meta-analysis. Model Fitting weight, d-index and confidence index (95%) of each research are shown in both figures. Although, the average outcome of most variables, except Dm BF, were estimated to be positive (Table 4), there are some studies that may in fact to be negative. Studies on the left side of the vertical line in Fig 2 and 3 presented negative values related to TMG. That is, the values found in these studies were lower than the values indicated by the TMG as normal values, while the

**Table 2. Egger regression, Begg and Mazumdar correlation test, and Rosenthal´s Fail-Safe Number.**

| Muscle | TMG variable | Egger regression Value (p-value) | B-M correlation Value (p-value) | R-F-S-N Value (p-value) |
|---|---|---|---|---|
| BF | Tc | −0.65 (0.51) | −0.18 (0.43) | 27 (<0.01) |
| | Dm | −0.74 (0.46) | −0,28 (0.36) | 13 (<0.01) |
| | Td | 0.30 (0.76) | 0.40 (0.48) | 0 (0.08) |
| ST | Dm | −0.73 (0.47) | 0.00 (1) | 2 (0.02) |
| | Td | −0.15 (0.88) | −0.33 (1) | 2 (0.02) |

B-M correlation = Begg and Mazumdar correlation test, R-F-S-N = Rosenthal´s Fail-Safe Number.

**Table 3. Heterogeneity statistics.**

| Muscle | TMG variable | I² | H² | Q | p-value |
|---|---|---|---|---|---|
| BF | Tc | 22.82% | 1.296 | 15.708 | 0.205 |
| | Dm | 0.01% | 1.00 | 7.489 | 0.485 |
| | Td | 0.03% | 1.00 | 5.194 | 0.268 |
| ST | Tc | 97.69% | 43.334 | 238.51 | < 0.001 |
| | Dm | 0% | 1.00 | 2.339 | 0.505 |
| | Td | 34.43% | 1.525 | 2.951 | 0.229 |

**Table 4. Random-effects model.**

| TMG Parameteres | k | $\breve{\mu}$ | CI (95%) Lower bound | CI (95%) Upper bound | Z | p |
|---|---|---|---|---|---|---|
| Tc BF | 13 | 0.144 | 0.036 | 0.251 | 2.61 | 0.009* |
| Dm BF | 9 | −0.155 | − 0.280 | − 0.029 | −2.41 | 0.015* |
| Td BF | 5 | 0.0913 | 0.014 | 0.197 | 1.69 | 0.90 |
| Dm ST | 4 | 0.230 | 0.013 | 0.446 | 2.08 | 0.038* |
| Td ST | 3 | 0.283 | 0.041 | 0.608 | 1.71 | 0.087 |

k = number of articles, $\breve{\mu}$ = Tau² Estimator: Restricted Maximum-Likelihood, CI = confidence interval.

**Table 5. Mixed-effects model.**

| TMG Parameteres | k | $\breve{\mu}$ | CI (95%) Lower bound | CI (95%) Upper bound | Z | p |
|---|---|---|---|---|---|---|
| Tc BF | 13 | 0.039 | -0.067 | 0.146 | 0.718 | 0.473 |
| Dm BF | 9 | 0.074 | − 0.088 | 0.236 | 0.892 | 0.373 |
| Td BF | 5 | −0.041 | −0.143 | 0.060 | −0.794 | 0.427 |
| Dm ST | 4 | 0.082 | −0.0274 | 0.191 | 1.468 | 0.142 |
| Td ST | 3 | −0.097 | −0.449 | 0.254 | −0.542 | −0.587 |

k = number of articles, $\breve{\mu}$ = Tau² Estimator: Restricted Maximum-Likelihood, CI = confidence interval.

values found in the studies which presented values on the right side of the vertical line, indicated higher values than those presented by the TMG as normal values. In Tc BF, the values of 9 of 13 included studies were higher than TMG values. Paravlic et al. [32] shows the largest model fitting weight in all variables included (Tc = 25.04%, Td = 73.66%) (Fig 2). In Dm BF, the average outcome was estimated to be negative, but one study was positive [33].Regarding to ST, all studies included show similar model fitting weights (Fig 3). The values in most of the included studies were higher than the TMG values.

According to the Cook's distances, none of the studies could be overly influential on any of the variables.

Table 6 shows descriptive values of muscles measured included in each research. All measured were made in dominant leg. Authors who select studies to be included in this meta-analysis, had an excellent inter-rater reliability (κ = 1).

Each group in each study were consider as an individual group in order to compare their values with the TMG values. In Table 7, weighted average values are presented. It can be observed that Tc in ST muscle was excluded from this analysis. And, consequently, Vrn has not been included either.

## Discussion

The aim of this review was twofold: 1) to examine the scientific evidence on TMG parameters of hamstring muscles in professional soccer players during the competitive season, and 2) to establish the most frequent values of hamstring muscles for the main TMG parameters in soccer players in comparison with the reference values provided by the TMG software. Likewise, the most-frequent values mean the most-repeated value of the contractile and mechanical properties of soccer players, but the most repeated value does not mean an adequate value. The main findings of the study are that of the six parameters analysed, three variables were found to differ significantly from the most frequent values of the TMG software and the values reported in the scientific evidence of the present review. The significant differences were in the Dm of both muscles: in the BF ($\leq 0.015$), and in the ST ($\leq 0.038$), and the Tc of the BF ($\leq 0.009$) (Table 4). However, no significant differences were found in the Td parameters of any muscle. Due to the heterogeneity of the results, it was not possible to perform the analysis of the ST contraction time, though all the values presented by the studies analysed are higher than the TMG Software values.

The time of contraction in milliseconds measured with TMG reflects the contraction velocity of the muscle. A muscle with a short time of contraction indicates high contraction velocity, and a muscle with a long time of contraction indicates a slow muscle [42]. Soccer players require muscles with fast contraction capabilities to accelerate and decelerate effectively. In the hamstrings muscles, a reduced Tc is desirable to minimize the risk of injury

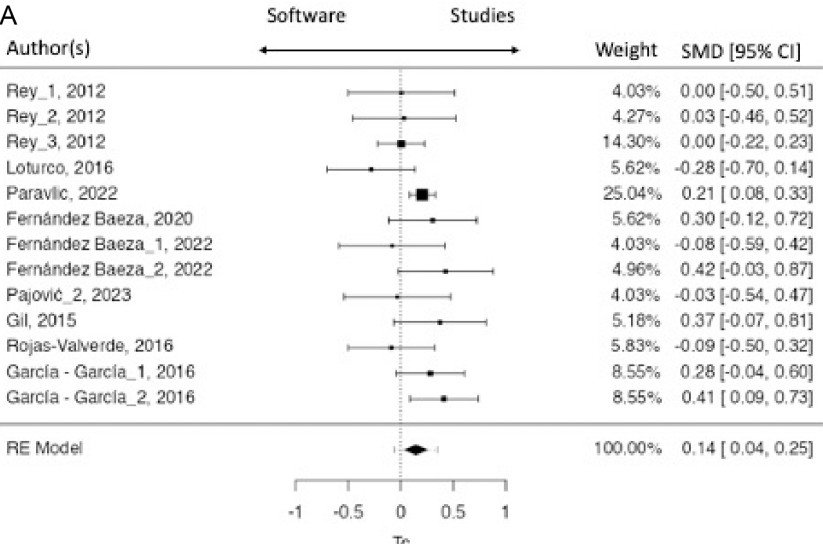

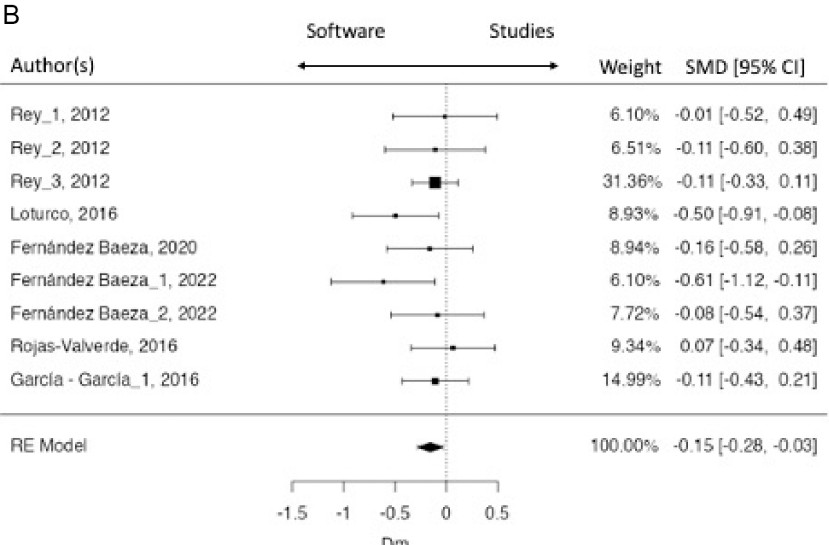

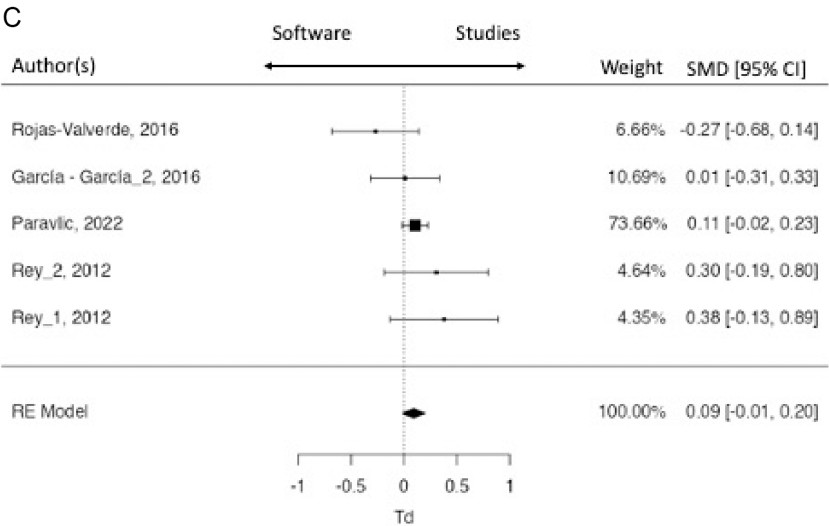

**Fig 2. BF´s Forest plot.**

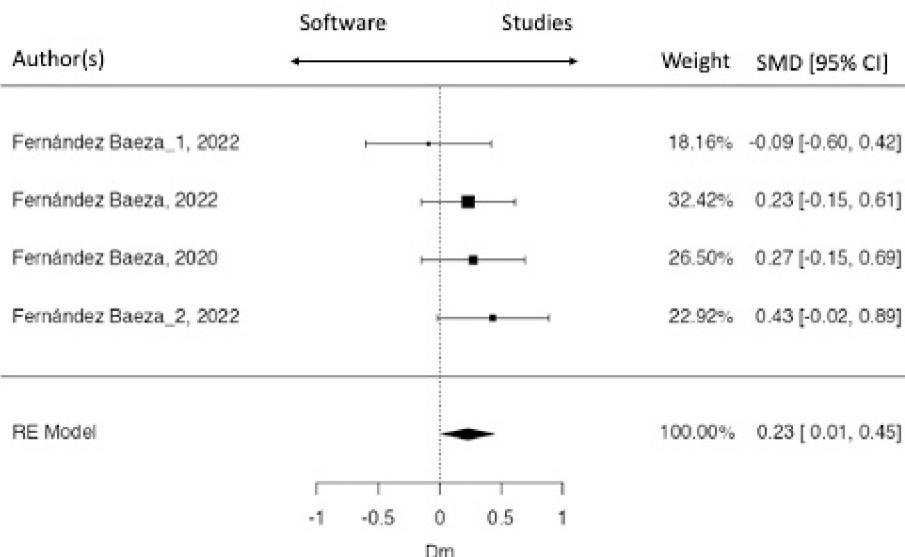

**Fig 3. ST´s Forest plot.**

**Table 6. Studies selected.**

| Code | References | Age (years) | Sample (n) | Muscles measured |
|------|-----------|-------------|-----------|------------------|
| Rey_1, 2012<br>Rey_2, 2012 | Rey et al., 2012 [34] | 23,5 ± 3,4 | 15 | BF dominant leg: Tc 26,7 ± 4,7 (ms); Dm 5,5 ± 1,7 (mm); Td 24,5 ± 1,9 (ms) |
| | | 23,5 ± 3,4 | 16 | BF dominant leg: Tc 26,9 ± 3,6 (ms); Dm 5,3 ± 1,8 (mm); Td 24,3 ± 1,3 (ms) |
| Rey_3, 2012 | Rey et al., 2012 [35] | 26,6 ± 4,4 | 78 | BF dominant leg: Tc 26.68 ± 4.45 (ms); Dm 5.30 ± 1.62 (mm), Td 24.72 ± 1.91 (ms), |
| Loturco, 2016 | Loturco et al., 2016 [36] | 23,8 ± 4,2 | 22 | BF dominant leg: Tc 24,5 ± 10,6 (ms); Dm 4,48 ± 1,99 (mm); Td 21,4 ± 2,2 (ms) |
| Fabok, 2019 | Fabok et al., 2019 [37] | 23 ± 4,4 | 54 | BF dominant leg: Tc 32,19 ± 7,64 (ms); Dm 7,39 ± 1,87 (mm); Td 25,56 ± 3,58 (ms) |
| Paravlic, 2022 | Paravlic et al., 2022 [32] | 22,59 ± 3,91 | 57 | BF dominant leg: Tc 28,25 ± 8,91 (ms); Dm 6,57 ± 2,73 (mm); Td 23,76 ± 2,94 |
| Fernández Baeza, 2022 | Fernández et al., 2022 [38] | 25,04 ± 4,5 | 27 | BF dominant leg: Tc 35.63 ± 10,97(ms); Dm 7.5 ± 2.23 (mm)<br>ST dominant leg: Tc 43,99 ± 7,81 (ms); 8,72 ± 2,21 (mm) |
| Fernández Baeza, 2020 | Fernández et al., 2020 [39] | 26,36 ± 4,68 | 22 | BF dominant leg: Tc 28,99 ± 6,94 (ms); Dm 5,19 ± 1,78 (mm); Tr 62,48 ± 27,37; Ts 191,29 ± 24,96<br>ST dominant leg: Tc 47,16 ± 9,06 (ms); Dm 8,84 ± 2,92 (mm); Tr 71,76 ± 25,72; Ts 149,87 ± 27,11 |
| Fernández Baeza_1, 2022<br>Fernández Baeza_2, 2022 | Fernández et al., 2022 [20] | 21,07 ± 1,73 | 15 | BF dominant leg: Tc 26,02 ± 8,54 (ms); Dm 4,23 ± 2,29 (mm)<br>ST dominant leg: Tc 48,27 ± 6,65 (ms); Dm 7,8 ± 2,11 (mm) |
| | | 26,68 ± 4,69 | 19 | BF dominant leg: Tc 29,92 ± 7,68 (ms); Dm 5,35 ± 1,39 (mm)<br>ST dominant leg: Tc 47,42 ± 9,05 (ms); Dm 9,3 ± 2,81 (mm) |
| Pajović, 2023<br>Pajović_1, 2023<br>Pajović_2, 2023 | Pajović et al., 2023 [40] | 24,09 ± 5,1 | 22 (defender) | BF dominant leg: Tc 19,5 ± 9,58 (ms); Dm 1,62 ± 1,41 (mm); Td 24,3 ± 24,5<br>ST dominant leg: Tc 44,9 ± 13,1 (ms); Dm 4,99 ± 2,9 (mm); Td 26,08 ± 7,13 |
| | | 26,8 ± 3,7 | 15 (midfielder) | BF dominant leg: Tc 26,4 ± 15,3 (ms); Dm 2,24 ± 1,53 (mm); Td 19,8 ± 5,17<br>ST dominant leg: Tc 46,5 ± 13,1 (ms); Dm 5,97 ± 2,51 (mm); Td 25,3 ± 3,78 |
| | | 25,5 ± 5,7 | 20 (attacker) | BF dominant leg: Tc 23,2 ± 11 (ms); Dm 2,77 ± 1,78 (mm); Td 20,05 ± 2,84<br>ST dominant leg: Tc 44,2 ± 13,01 (ms); Dm 5,07 ± 2,42 (mm); Td 24,3 ± 4,44 |
| Gil, 2015 | Gil et., 2015 [41] | 23,3 ± 4,8 | 20 | BF dominant leg: Tc 29,53 ± 11,09 (ms); Dm7,18 ± 3,39 (mm) |
| García García_1, 2016 | García et., 2016 [42] | 27,2 ± 3,3 | 37 | BF dominant leg: Tc 28,8 ± 5,9 (ms); Dm 5,3 ± 1,3 (mm); Td 21 ± 1,6 (ms); |
| García García_2, 2016 | | 27,2 ± 3,3 | 37 | BF dominant leg: Tc 29,8 ± 4,6 (ms); Dm 6,6 ± 1,9 (mm); Td 23.5 ±1,3 (ms) |
| Rojas-Valverde, 2016 | Rojas-Valverde et., 2016 [33] | 24,78 ± 3,9 | 23 | BF Dominant leg: Tc 25,96 ± 6,37 (ms); Dm 5,67 ± 1,88 (ms); Td 22,75 ± 1,92 (ms) |

**Table 7. Weighted average values in the studies analysed and TMG Software of TMG parameters.**

| | | Tc (ms) Mean ± SD | n | Dm (mm) Mean ± SD | n | Td (ms) Mean ± SD | n | Vrn (mm/s) |
|---|---|---|---|---|---|---|---|---|
| Reviewed articles | BF | 27.88 ± 7.68 | 585 | 5.2 ± 1.68 | 247 | 23.72 ± 2.59 | 357 | 0.0287 |
| | ST | – | – | 8.72 ± 2.52 | 83 | 25.25 ± 5.30 | 57 | – |
| TMG software | BF | 26.66 ± 7.67 | 5511 | 5.53 ± 2.12 | 5511 | 23.47 ± 2.71 | 5511 | 0.03 |
| | ST | 40.77 ± 1.06 | 1548 | 8.06 ± 2.86 | 1548 | 24.30 ± 3.25 | 1548 | 0.0196 |

during high-intensity movements. The Tc of the BF are slower in the results of the present review (27.88 ± 7.65 ms) than in the TMG software (26.66 ± 7.67 ms). These results are similar (27.23 ms) in elite soccer players in the study by Paravlic. [30] who compared different levels of players, being less trained players (41.88 ms). Furthermore, in studies of the present review that measured ST, the Tc is slower in all the studies of the present review than in the TMG software. This may be due to the fact that the selected studies are measurements in the competitive season and the software data have been measured in pre-season, as it has been observed that the contraction time increases with fatigue, which is caused by training and matches [27]. Moreover, the ST has the slowest contraction time of all the muscles in a soccer player's thigh (41.44 ms) [30]. Muscular displacement, measured by TMG, is a parameter that reflects muscle tone and stiffness [26]. If the stimulated muscle has great displacement, it indicates a lack of muscular tone, and when the muscle has little displacement, it shows stiffness in the muscle. On the contrary, if the muscle's displacement is adequate, it demonstrates good muscle tone. An adequate Dm in the hamstring muscles allows for a greater range of motion and energy absorption capacity during sprints and jumps, which can reduce the risk of injuries related to excessive muscle strain. The Dm of the BF is higher in the TMG software (5.53 ± 2.12 mm) than the results of the present review (5.2 ± 1.68 mm) (Table 7). This may be because the players in the review sample are professional players and have better muscle tone than the players in the TMG software sample. Paravlic's study [30] found

that as the player`s level increased, the Dm decreased: elite players had a Dm of 4.97 mm, highly trained players had 5.25 mm, and trained players had 7.71 mm. The value of the Dm of the ST is higher in the studies analysed than in the TMG software. This may be due to the fact that the studies have been measured in competitive season and the TMG software has been measured in pre-season, and the ST is worsening throughout the competitive season the muscular tone [20]. Likewise, the ST is the muscle with the worst muscle tone in all the studies analysed and in the TMG software.

Regarding the interaction between the BF and ST, the data for the two muscles are very different: the Tc measured in milliseconds was 26.66 ± 7.67 in BF and 40.77 ± 1.06 in ST (Table 7), the Dm measured in millimeters was 5.53 ± 2.12 in the BF and 8.06 ± 2.86 in the ST (Table 7), and the Vrn measured in mm/s was 0.03 in the BF and 0.0196 in the ST (Table 7). These differences in the three variables in these synergistic muscles, which should have a certain coordination, make the BF work harder due to the lack of muscle tone of the ST [22]. Moreover, the BF is the most injured muscle of the hamstring's muscles [17]. One reason for this is because of the increase of the high intensity running and sprinting in a match in recent seasons in soccer [43]. This has possibly increased hamstring injuries in recent years and is now the most common injury in soccer [37]. The study by Wilmes et al. [44] observed that in high-speed sprinting, the BF has a greater activation than the ST, and that both muscles have different activation times [45]. Moreover, the BF fatigues more than the ST [46]. Furthermore, the TMG values of Dm and Tc between the BF and ST evolve in opposite ways during the competitive season. The BF becomes a muscle that improves both parameters by

reducing Tc and Dm; however, the ST worsens its values by becoming a slow muscle with less muscle tone [38].

Therefore, to reduce the hamstring risk of injury in the BF, different strategies that have been found to reduce the risk of hamstring injury can be used with players, such as eccentric training [47], the use of an adequate dose of sprint in training session [48], and designing an individualized training programme for each player based on TMG data [20], since in this study it was observed that this type of training decreases the differences between ST and BF, reducing the number of BF injuries.

The present review is limited by the number of studies analysed. There are parameters that show significant differences between the software and the studies analysed, but the number of players measured in the TMG software is higher than in the studies analysed. In addition, the semitendinosus is an under-measured muscle compared to the BF. Further studies are needed in order to determine the most common values in professional soccer players in competitive season, especially of the ST muscle. To conclude, another limitation of the present study lies in the potential bias in the selection of TMG parameters. The analysis prioritized Dm and Tc, as these are the most commonly reported parameters in the scientific literature, facilitating their interpretation and comparison with previous studies. However, this choice may have excluded relevant information provided by Tr and Ts, parameters that, although less studied and potentially more sensitive to fatigue or sensor positioning, offer a complementary perspective on muscle dynamics. This limitation could also be influenced by the lower consistency in measuring these latter parameters. Future research should consider a more inclusive approach that incorporates all parameters to provide a more comprehensive assessment of muscle behavior.

## Conclusions

The software values describe the reality of the contractile and mechanical properties of the hamstrings muscles in football players, as there is no great difference with the studies analysed. Therefore, the TMG is presented as a useful device to measure the contractile and mechanical properties and determine the values that occur most frequently, however these values do not mean that they are adequate values, because the sport itself produces muscular imbalances, which must be corrected in training. In addition, the most frequent BF and ST values of soccer players are very different in the TMG parameters between the two synergistic muscles. Likewise, the ST is the slowest muscle and lacks muscle tone in hamstrings muscles. Moreover, the study shows that the BF is a muscle that gets very fatigued due to matches and training.

## Supporting information

**S1 File. The Newcastle–Ottawa scale (NOS) scoring.**
(DOCX)

**S1 Checklist. PRISMA 2020 checklist.**
(DOCX)

## Author contributions

**Conceptualization:** Daniel Fernández-Baeza, Cristina González-Millán.

**Data curation:** Germán Díaz-Ureña.

**Formal analysis:** Daniel Fernández-Baeza, Cristina González-Millán.

**Investigation:** Daniel Fernández-Baeza.

**Methodology:** Daniel Fernández-Baeza, Germán Díaz-Ureña.

**Supervision:** Cristina González-Millán.

**Validation:** Germán Díaz-Ureña.

**Visualization:** Cristina González-Millán.

**Writing – original draft:** Daniel Fernández-Baeza.

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
