## [Decision Letter · Decision Letter 0]

20 Aug 2024

PONE-D-24-11496Muscle contractile properties of hamstrings in professional soccer players: A systematic review and meta-analysisPLOS ONE

Dear Dr. FERNANDEZ BAEZA,

Thank you for submitting your manuscript to PLOS ONE. After careful consideration, we feel that it has merit but does not fully meet PLOS ONE’s publication criteria as it currently stands. Therefore, we invite you to submit a revised version of the manuscript that addresses the points raised during the review process.

We look forward to receiving your revised manuscript.

Kind regards,

Holakoo Mohsenifar

Academic Editor

PLOS ONE

2. We note that your Data Availability Statement is currently as follows: [All relevant data are within the manuscript and its Supporting Information files]

Reviewers' comments:

Reviewer's Responses to Questions

**Comments to the Author**

1. Is the manuscript technically sound, and do the data support the conclusions?

Reviewer #1: Partly

Reviewer #2: Yes

Reviewer #3: Partly

2. Has the statistical analysis been performed appropriately and rigorously? 

Reviewer #1: Yes

Reviewer #2: Yes

Reviewer #3: No

3. Have the authors made all data underlying the findings in their manuscript fully available?

Reviewer #1: Yes

Reviewer #2: Yes

Reviewer #3: Yes

4. Is the manuscript presented in an intelligible fashion and written in standard English?

Reviewer #1: No

Reviewer #2: Yes

Reviewer #3: No

5. Review Comments to the Author

Reviewer #1: 1.In the Abstract section, the results of this review you demonstrated (”The main findings of the study are that of the six parameters analysed, three variables were found to differ significantly.”) can not solve the two purpose you proposed at the beginning of this “Abstract” .Please make it clear and detailed.

2.In the introduction section, just like you mentioned in the title, you want to know the scientific evidence of hamstring TMG parameters and establish the most-frequent values of hamstring for the main TMG parameters. We all know that several muscles involved in the soccer process, why you emphasize on the hamstring? Please state the necessity relationship, or importance of this muscle on soccer players.

3.In terms of the search strategy section, why didn’t you search other important databases, like cochrane library(CENTRAL), EMBASE, PEDro, et al. ?

4.Line 75, Inclusion criteria were male, why? Please clarify the reason why women were excluded.

Reviewer #2: The authors must be commended for carrying out an excellent review study on contractile and mechanical properties of hamstring muscles measured by the method of tensiomyography in professional soccer players. This topic is important, interesting, and novel, the methodology and study design used in the study are appropriate and the manuscript is written with high clarity. However, some issues need to be taken into consideration.

Title

- I strongly suggest modifying the title so it can be more descriptive. For example: Contractile and mechanical properties of hamstring muscles measured by the method of tensiomyography (TMG) in professional soccer players: A systematic review and meta-analysis

Abstract

- Add a statistical analysis that you used in the study.

Introduction

- Line 19: ’’...for measuring muscle mechanical characteristics.’’. I suggest adding - and contractile, since TMG is measuring not only mechanical properties.

- I suggest briefly explaining what exactly TMG is used for. TMG is used to measure muscle contraction velocity and stiffness in the first place. Also, TMG is not used just for the assessment of muscle fatigue, but, more importantly, for the assessment of functional and lateral asymmetry. And finally, the aim of TMG diagnostics – injury prevention, sports performance, and rehabilitation monitoring.

- Im not sure what was the criteria for presenting the previous studies. I strongly suggest revising the second and third paragraphs (which you can merge to one), presenting the previous studies more clearly.

- Line 56 and 57: Please add MUSCLES when you are referring to hamstring muscles, not just hamstring.

Methods

- Search strategy: What about exclusion criteria? Please elaborate.

Study selection and data extraction: Why did you extract only these three parameters? Please elaborate.

Study selection and data extraction: Why did you extract only these two muscles? Please elaborate.

Results

- Move Figure 1 to the methods section.

- Please revise the Table 5. For example, you emphasized that in the study Pajovic et al., 2023. the sample was conducted by defenders, which is not the case.

Discussion

- First paragraph, second sentence: Please emphasize at the end of this kind of sentence a place in the results where the reader can see this information (e.g. Table 2 or 3).

- Line 164: I strongly suggest replacing explosiveness with contraction velocity or speed of force twitch generation.

Reviewer #3: Comments on:

“Muscle contractile properties of hamstrings in professional soccer players: A

systematic review and meta-analysis”

First, I would like to congratulate the authors on selecting a topic that holds significant potential. I believe this study can effectively inform football practitioners about TMG normative values for hamstring muscles, which may impact training optimization. In the submitted article, the authors aimed to: 1) gather scientific evidence on hamstring TMG parameters in professional football players during the season, and 2) establish the most frequent values for the main TMG parameters in soccer players, comparing these with the reference values provided by the TMG software. The results showed slight but significant discrepancies between the reference values from the TMG software and those identified in the literature. While I find this article very interesting for readers, several important issues must be addressed before it can be accepted for publication.

General comments:

- The introduction section lacks a clear problem identification based on previously published literature investigating similar topics. Proper problem identification is crucial, as it provides a solid foundation for why the study is needed. When constructing the narrative, the authors frequently jump from one topic to another, often unrelated to their main aim. By the end of the introduction, it feels as though they abruptly shift from the rest of the introduction to discussing the limitations of current literature—an issue not previously addressed. This is particularly confusing for readers trying to understand the background and the rationale behind comparing TMG reference values provided by the software with those found in published literature.

- In the introduction section, there are several references to the usefulness of TMG as a device that can aid in injury prevention (e.g., Lines 29, 32, 44). While a theoretical framework suggesting that muscle contractile properties may be a risk factor for subsequent injury has been proposed (see references: https://pubmed.ncbi.nlm.nih.gov/34515974/, https://doi.org/10.5114/biolsport.2025.139853, https://doi.org/10.3390/ijerph19010474), there is no strong evidence from prospective studies utilizing TMG to investigate this question. Therefore, I suggest being less definitive in the introduction regarding this issue. Instead, the introduction should provide stronger evidence from existing literature to support any assumptions regarding TMG's role in injury prevention.

- Throughout the manuscript, there is a noticeable mismatch between statements and the references used to support them. For example:

• On page 3, lines 43-44: The manuscript states that Faude et al. [22] investigated the impact of neuromuscular fatigue on sprint and jump performance in elite youth soccer players, emphasizing the importance of monitoring muscle function to optimize training and prevent injuries. However, Faude et al. did not assess the impact of neuromuscular fatigue on sprint and jump performance in that publication.

• On page 3, lines 46-48: The reference to Ekstrand et al. [24] does not support the identification of hamstring injury mechanisms related to muscle imbalances; instead, it discusses injury burden and differences in injury incidence among various hamstring muscles.

To strengthen your narrative, please refer to more appropriate references that accurately support the claims made.

- When the authors refer to “during a season,” do they mean the competitive season? While the discussion section mentions the competitive season (Line 168), this should be clarified earlier in the manuscript. Please specify this in the methods section and provide a rationale for why this particular period was chosen for TMG data aggregation.

• The authors' decision to compare the results of the summarized literature with the reference values provided by the software is interesting idea. However, this approach necessitates more complex models including normalization of the published values according to participants’ age, playing position, and playing status, as these factors have been shown to influence TMG parameters in athletes (https://doi.org/10.3390/ijerph20020924, https://doi.org/10.3390/sports10110177, https://doi.org/10.5114/biolsport.2025.139853). Also, more insights about the reference values provided by TMG software should be presented for better interpretation of current findings. Authors mentioned in the subchapter “Statistical analyses” that “the control group mean, and standard deviation were weighted using a sample size of forward, centre, and defense soccer players creating a single group. I believe more clarification about the control group should be provided here.

- A similar article was recently published (https://doi.org/10.5114/biolsport.2025.139853) that provided summarized TMG values for all lower limb muscles and parameters in soccer players available in the literature. The findings from this published article can serve as a valuable reference point to strengthen your introduction and discussion sections, which currently lack strong arguments for both problem identification and the interpretation of the results observed.

- The limitation section warrants further improvements as there are several issues with the authors approach and statistical models used to compare reference values, that must be addressed in detail.

Minor comments:

- Page 2, Line 22-23: The sentence starting with “A few years ago” is out of context and does not belong here. Please consider removing or rephrasing it.

- Page 3, Line 23: Consider using the term "device" instead of "tool" for consistency. Ensure that the chosen term is used consistently throughout the manuscript.

- Page 3, Line 23-26: I suggest shortening the sentence that starts with “Nowadays, …” to improve clarity and readability.

- Page 3, Line 27: While soccer is the most popular sport, it is not necessarily the most demanding, either physically or psychologically. Please consider removing this statement.

- Page 3, Line 29-30: Reference 15 does not support the claim about the existence of muscular imbalances in soccer. There are several other instances where references do not support the corresponding statements. Please review the entire manuscript carefully to address this issue.

- Page 3, Line 36: Provide more details on the results of the study by Piqueras-Sanchiz et al. This will offer readers better insights into previous studies and highlight areas that warrant further investigation.

- Page 3, Line 39: Muscle activation is not something that can be directly assessed by TMG, although it is sometimes interpreted as such. Please be more precise here, as TMG primarily provides insights into muscle contractile (i.e., mechanical) properties.

- From the authors' perspective, how would you differentiate between muscle activation and contractile response? Please clarify this distinction.

- Page 3, Line 40: What is the difference between asymmetries and imbalances in this context? Be more precise, as these terms could be interpreted as interchangeable, though it seems the authors intend to convey different concepts.

6. PLOS authors have the option to publish the peer review history of their article (what does this mean? ). If published, this will include your full peer review and any attached files.

**Do you want your identity to be public for this peer review?** For information about this choice, including consent withdrawal, please see our Privacy Policy .

Reviewer #1: No

Reviewer #2: No

Reviewer #3: **Yes: ** Armin H. Paravlic

---

## [Author Response · Author response to Decision Letter 1]

30 Sep 2024

Reviewer #1:

In the Abstract section, the results of this review you demonstrated (”The main findings of the study are that of the six parameters analysed, three variables were found to differ significantly cannot solve the two purpose you proposed at the beginning of this “Abstract”. Please make it clear and detailed

- Reply, added: “Furthermore, the weighted mean values founded were biceps femoris (Tc 27.88, Dm 5.2, Td 23.72) and in semitendinosus (Dm 8.72, Td 25.25)”.

2.In the introduction section, just like you mentioned in the title, you want to know the scientific evidence of hamstring TMG parameters and establish the most-frequent values of hamstring for the main TMG parameters. We all know that several muscles involved in the soccer process, why you emphasize on the hamstring? Please state the necessity relationship, or importance of this muscle on soccer players.

- Reply: following the considerations of a reviewer, the introduction section has been rewritten completely. Adding a new reference “In recent years hamstring injuries in soccer have increased by 4% [17]. In a longitudinal study from 2001 - 2022, hamstring injuries constitute 24% of all injuries in elite UEFA teams [18]”.

3.In terms of the search strategy section, why didn’t you search other important databases, like cochrane library (CENTRAL), EMBASE, PEDro, et al.?

- Reply: Thanks so much for your suggestion. We will keep that in mine for the next time.

4.Line 75, Inclusion criteria were male, why? Please clarify the reason why women were excluded.

- Reply, added “Women football players were not included due to the significant differences in physical, physiological, and biomechanical characteristics between male and female football players. Since the focus of the study is to homogenize the sample to more precisely analyze the impact of these variables on male football players, including women could introduce biases that would compromise the internal validity of the findings”.

Reviewer #2:

The authors must be commended for carrying out an excellent review study on contractile and mechanical properties of hamstring muscles measured by the method of tensiomyography in professional soccer players. This topic is important, interesting, and novel, the methodology and study design used in the study are appropriate and the manuscript is written with high clarity. However, some issues need to be taken into consideration.

Title

- I strongly suggest modifying the title so it can be more descriptive. For example: Contractile and mechanical properties of hamstring muscles measured by the method of tensiomyography (TMG) in professional soccer players: A systematic review and meta-analysis and metaregression

- Reply: Title changed

Abstract

- Add a statistical analysis that you used in the study.

- Reply, added: Study results were analyzed using restricted maximum-likelihood and random-effects models

Introduction

- Line 19: ’’...for measuring muscle mechanical characteristics.’’. I suggest adding - and contractile, since TMG is measuring not only mechanical properties.

- Following the considerations of a reviewer, the introduction section has been rewritten completely.

- Reply, added mechanical and contractile properties in all the documents.

- I suggest briefly explaining what exactly TMG is used for. TMG is used to measure muscle contraction velocity and stiffness in the first place. Also, TMG is not used just for the assessment of muscle fatigue, but, more importantly, for the assessment of functional and lateral asymmetry. And finally, the aim of TMG diagnostics – injury prevention, sports performance, and rehabilitation monitoring.

- Reply, following the considerations of a reviewer, the introduction section has been rewritten completely. The new version explains the different uses of TMG more clearly:

- Im not sure what was the criteria for presenting the previous studies. I strongly suggest revising the second and third paragraphs (which you can merge to one), presenting the previous studies more clearly.

- Reply, following the considerations of a reviewer, the introduction section has been rewritten completely.

- Line 56 and 57: Please add MUSCLES when you are referring to hamstring muscles, not just hamstring.

- Reply, added hamstring muscles in all the document

Methods

- Search strategy: What about exclusion criteria? Please elaborate.

- Reply, added: “Women football players were not inluded due to the significant differences in physical, physiological, and biomechanical characteristics between male and female football players. Since the focus of the study is to homogenize the sample to more precisely analyze the impact of these variables on male football players, including women could introduce biases that would compromise the internal validity of the findings”. Studies that reported an inappropriate population for this review were excluded. Studies with a sample of teenagers and players with more than 29 years old were excluded

Study selection and data extraction: Why did you extract only these three parameters? Please elaborate.

- Reply, ADDED: “The TMG variables extracted were three: contraction time (Tc), this parameter is fundamental for analyzing how quickly a muscle can generate force, directly influencing performance. Delay time (Td) because this is crucial for understanding muscle activation speed, which is essential in sports, where reaction time is key. Muscle displacement (Dm) a key indicator of its contraction and recovery capacity, it is relevant for monitoring muscle fatigue and assessing the risk of injury. These three parameters (Td, Tc, Dm) allows for a comprehensive evaluation of muscle performance and injury prevention, aligning directly with the study's objectives in elite athletes. The relaxion time (Tr) and sustain time (Ts) variables were not extracted because, while complementary, they are not as crucial for evaluating the central aspects of muscle performance that are of greatest interest in this study.

Study selection and data extraction: Why did you extract only these two muscles? Please elaborate.

- Reply ADDED: “the TMG- parameters of the knee flexor muscles: biceps femoris (BF) and semitendinosus (ST)). These two muscles provide a more specific and relevant assessment of the contractile and mechanical properties that directly impact athletic performance and injury prevention”.

Reply: TMG cannot measure semimembranosus muscle. TMG only can measure superficial muscles.

Results

- Move Figure 1 to the methods section.

- Reply: We follow PRISMA statement “The item 16a Result – study selection: Describe the results of the search and selection process, from the number of records identified in the search to the number of studies included in the review, ideally using a flow diagram”

- Please revise the Table 5. For example, you emphasized that in the study Pajovic et al., 2023. the sample was conducted by defenders, which is not the case.

- Reply: In the Pajovic et al. study, in methods, it can be read: “The sample participants consisted of 57 football players divided into three groups: defenders—DF (N = 22, Age = 24.09 ± 5.1 years, BW = 82.09 ± 7.07 kg, BH = 186.4 ± 5.6 cm), midfielders—MF (N = 15, Age = 26.8 ± 3.7 years, BW = 79.06 ± 7.2 kg, BH = 182.2 ± 6.5 cm), and forwards—FW (N = 20, Age = 25.5 ± 5.7 years, BW = 79.6 ± 5.2 kg, BH = 183.5 ± 5.2 cm)”. If you know any mistake please let us know. https://www.ncbi.nlm.nih.gov/pmc/articles/PMC9859018/

Discussion

- First paragraph, second sentence: Please emphasize at the end of this kind of sentence a place in the results where the reader can see this information (e.g. Table 2 or 3).

- Reply, Added

- Line 164: I strongly suggest replacing explosiveness with contraction velocity or speed of force twitch generation.

- Reply Changed: “The time of contraction in milliseconds measured with TMG reflects the contraction velocity of the muscle. A muscle with a short time of contraction indicates high contraction velocity”

RESPONSE REVIWER #3:

General comments:

- The introduction section lacks a clear problem identification based on previously published literature investigating similar topics. Proper problem identification is crucial, as it provides a solid foundation for why the study is needed. When constructing the narrative, the authors frequently jump from one topic to another, often unrelated to their main aim. By the end of the introduction, it feels as though they abruptly shift from the rest of the introduction to discussing the limitations of current literature—an issue not previously addressed. This is particularly confusing for readers trying to understand the background and the rationale behind comparing TMG reference values provided by the software with those found in published literature.

- Reply, following your considerations, the introduction section has been rewritten completely.

- In the introduction section, there are several references to the usefulness of TMG as a device that can aid in injury prevention (e.g., Lines 29, 32, 44). While a theoretical framework suggesting that muscle contractile properties may be a risk factor for subsequent injury has been proposed (see references: https://pubmed.ncbi.nlm.nih.gov/34515974/, https://doi.org/10.5114/biolsport.2025.139853, https://doi.org/10.3390/ijerph19010474), there is no strong evidence from prospective studies utilizing TMG to investigate this question. Therefore, I suggest being less definitive in the introduction regarding this issue. Instead, the introduction should provide stronger evidence from existing literature to support any assumptions regarding TMG's role in injury prevention.

- Reply, following your considerations, the introduction section has been rewritten completely.

- Throughout the manuscript, there is a noticeable mismatch between statements and the references used to support them. For example:

• On page 3, lines 43-44: The manuscript states that Faude et al. [22] investigated the impact of neuromuscular fatigue on sprint and jump performance in elite youth soccer players, emphasizing the importance of monitoring muscle function to optimize training and prevent injuries. However, Faude et al. did not assess the impact of neuromuscular fatigue on sprint and jump performance in that publication.

- Reply, following your considerations, the introduction section has been rewritten completely.

- Reply: Removed

• On page 3, lines 46-48: The reference to Ekstrand et al. [24] does not support the identification of hamstring injury mechanisms related to muscle imbalances; instead, it discusses injury burden and differences in injury incidence among various hamstring muscles. To strengthen your narrative, please refer to more appropriate references that accurately support the claims made.

- Reply, following your considerations, the introduction section has been rewritten completely.

- Reply: Removed

When the authors refer to “during a season,” do they mean the competitive season? While the discussion section mentions the competitive season (Line 168), this should be clarified earlier in the manuscript. Please specify this in the methods section and provide a rationale for why this particular period was chosen for TMG data aggregation.

- Reply: Clarified in all the document with “competitive season”.

• The authors' decision to compare the results of the summarized literature with the reference values provided by the software is interesting idea. However, this approach necessitates more complex models including normalization of the published values according to participants’ age, playing position, and playing status, as these factors have been shown to influence TMG parameters in athletes (https://doi.org/10.3390/ijerph20020924, https://doi.org/10.3390/sports10110177, https://doi.org/10.5114/biolsport.2025.139853). Also, more insights about the reference values provided by TMG software should be presented for better interpretation of current findings.

- Reply: A meta-regression model included age were performed. Playing position and player status were not included due to the characteristics of the sample. TMG studies do not distinguish by positions in most cases. We had assumed that the playing status referred to health status like in Paravlic (2024). (Establishing reference values for tensiomyography-derived parameters in soccer players: insights from a systematic review, meta-analysis and meta-regression). All our sample were healthy status.

Authors mentioned in the subchapter “Statistical analyses” that “the control group mean, and standard deviation were weighted using a sample size of forward, centre, and defence soccer players creating a single group. I believe more clarification about the control group should be provided here.

- Reply: changed the text related to the control group “The mean value and standard deviation of the control group were calculated from the weighted mean value and standard deviation based on the sample size of the three subgroups from the TMG software”

- A similar article was recently published (https://doi.org/10.5114/biolsport.2025.139853) that provided summarized TMG values for all lower limb muscles and parameters in soccer players available in the literature. The findings from this published article can serve as a valuable reference point to strengthen your introduction and discussion sections, which currently lack strong arguments for both problem identification and the interpretation of the results observed.

- Reply ADDED many times this reference, in the introduction and discussion sections.

- The limitation section warrants further improvements as there are several issues with the authors approach and statistical models used to compare reference values, that must be addressed in detail.

- Reply we added a meta-regression model in the methods and results. If you consider any other comments please let us know.

Minor comments:

- Page 2, Line 22-23: The sentence starting with “A few years ago” is out of context and does not belong here. Please consider removing or rephrasing it.

- - Reply, following your considerations, the introduction section has been rewritten completely.

- Reply: Removed

- Page 3, Line 23: Consider using the term "device" instead of "tool" for consistency. Ensure that the chosen term is used consistently throughout the manuscript.

- Reply Changed in all the document

- Page 3, Line 23-26: I suggest shortening the sentence that starts with “Nowadays, …” to improve clarity and readability.

- Reply, following your considerations, the introduction section has been rewritten completely.

- Reply: Removed

- Page 3, Line 27: While soccer is the most popular sport, it is not necessarily the most demanding, either physically or psychologically. Please consider removing this statement.

- Reply Removed

- Page 3, Line 29-30: Reference 15 does not support the claim about the existence of muscular imbalances in soccer.

- Reply, following your considerations, the introduction section has been rewritten completely.

- Reply: Removed

There are several other instances where references do not support the corresponding statements. Please review the entire manuscript carefully to address this issue.

-Reply Corrected in all the document

- Page 3, Line 36: Provide more details on the results of the study by Piqueras-Sanchiz et al. This will offer readers better insights into previous studies and highlight areas that warrant further investigation.

- Reply, following your considerations, the introduction section has been rewritten completely.

- Reply: Removed

- Page 3, Line 39: Muscle activation is not something that can be directly assessed by TMG, although it is sometimes interpreted as such. Please be more precise here, as TMG primarily provides insights into muscle contractile (i.e., mechanical) properties. - From the authors' perspective, how would you differentiate between muscle activation and contractile response? Please clarify this distinction.

- Reply, removed

- Page 3, Line 40: What is the difference between asymmetries and imbalances in this context? Be more precise, as these terms could be interpreted as interchangeable, though it seems the authors int

---

## [Decision Letter · Decision Letter 1]

21 Nov 2024

PONE-D-24-11496R1Contractile and mechanical properties of hamstring muscles measured by the method of tensiomyography (TMG) in professional soccer players: A systematic review, meta-analysis and meta-regressionPLOS ONE

Dear Dr. FERNANDEZ BAEZA,

Thank you for submitting your manuscript to PLOS ONE. After careful consideration, we feel that it has merit but does not fully meet PLOS ONE’s publication criteria as it currently stands. Therefore, we invite you to submit a revised version of the manuscript that addresses the points raised during the review process.

We look forward to receiving your revised manuscript.

Kind regards,

Holakoo Mohsenifar

Academic Editor

PLOS ONE

Journal Requirements:

Reviewers' comments:

Reviewer's Responses to Questions

**Comments to the Author**

1. If the authors have adequately addressed your comments raised in a previous round of review and you feel that this manuscript is now acceptable for publication, you may indicate that here to bypass the “Comments to the Author” section, enter your conflict of interest statement in the “Confidential to Editor” section, and submit your "Accept" recommendation.

Reviewer #4: All comments have been addressed

2. Is the manuscript technically sound, and do the data support the conclusions?

Reviewer #4: Yes

3. Has the statistical analysis been performed appropriately and rigorously? 

Reviewer #4: Yes

4. Have the authors made all data underlying the findings in their manuscript fully available?

Reviewer #4: Yes

5. Is the manuscript presented in an intelligible fashion and written in standard English?

Reviewer #4: Yes

6. Review Comments to the Author

Reviewer #4: - The introduction section addresses the relevance of TMG and its utility in evaluating muscle properties. However, it lacks a structured approach and problem identification. The rationale behind focusing on hamstring muscles in soccer is not fully developed. To enhance the introduction:

- Clarify the Problem Statement: Elaborate on the importance of TMG in evaluating specific parameters critical to injury prevention, especially for soccer players prone to hamstring injuries.

- Focus on Existing Gaps: The transition from general TMG applications to the study's objectives is abrupt. Consider discussing gaps in previous literature regarding TMG's efficacy for muscle-specific injury risk assessment.

- Discussion: Comparison with Existing Literature: The manuscript could improve by systematically comparing its findings with previous literature to contextualize the significance of specific TMG parameters.

Interpretation of Tc, Td, and Dm Differences: The physiological interpretation of contraction time (Tc), muscle displacement (Dm), and delay time (Td) is crucial but underexplored. Explaining how these parameters relate to hamstring function and soccer-specific demands could enhance relevance.

Limitations: While some limitations are mentioned, the manuscript could expand on them, especially regarding the small sample sizes for certain parameters and the exclusion of Tr and Ts values. Additionally, the limited number of studies on the semitendinosus muscle and potential biases in TMG parameter selection should be acknowledged.

- Minor typographical errors in the results and discussion sections should be corrected.

This study presents a valuable contribution to the field of sports science and injury prevention in professional soccer. However, refinements to the methodology, expanded discussion on limitations, and improved adherence to reporting guidelines are recommended to elevate the quality and impact of the manuscript

7. PLOS authors have the option to publish the peer review history of their article (what does this mean? ). If published, this will include your full peer review and any attached files.

**Do you want your identity to be public for this peer review?** For information about this choice, including consent withdrawal, please see our Privacy Policy .

Reviewer #4: **Yes: ** Esedullah AKARAS

---

## [Author Response · Author response to Decision Letter 2]

24 Nov 2024

The introduction section addresses the relevance of TMG and its utility in evaluating muscle properties. However, it lacks a structured approach and problem identification. The rationale behind focusing on hamstring muscles in soccer is not fully developed. To enhance the introduction:

- Clarify the Problem Statement: Elaborate on the importance of TMG in evaluating specific parameters critical to injury prevention, especially for soccer players prone to hamstring injuries.

Response: To clarify, it is important to note that, to date, there is only one experimental study in which an individualized training program was implemented with the specific aim of preventing injuries. This highlights the need for further research on the effectiveness of TMG in assessing key parameters that could aid in injury prevention, particularly for soccer players who are prone to hamstring injuries.

- Focus on Existing Gaps: The transition from general TMG applications to the study's objectives is abrupt. Consider discussing gaps in previous literature regarding TMG's efficacy for muscle-specific injury risk assessment.

Response: Enhanced the final segment of the introduction by incorporating additional details to better substantiate the justification of the research problem, providing a more compelling context before transitioning to the objectives.

- Discussion: Comparison with Existing Literature: The manuscript could improve by systematically comparing its findings with previous literature to contextualize the significance of specific TMG parameters. Interpretation of Tc, Td, and Dm Differences: The physiological interpretation of contraction time (Tc), muscle displacement (Dm), and delay time (Td) is crucial but underexplored. Explaining how these parameters relate to hamstring function and soccer-specific demands could enhance relevance.

Response: In the Study Selection and data extraction section, this required issue is explained a bit more. We added more information about Td in this section. Additionally, in the Discussion, we have added more information about the Tc and Dm.

Limitations: While some limitations are mentioned, the manuscript could expand on them, especially regarding the small sample sizes for certain parameters and the exclusion of Tr and Ts values. Additionally, the limited number of studies on the semitendinosus muscle and potential biases in TMG parameter selection should be acknowledged.

Response: added the suggestions

- Minor typographical errors in the results and discussion sections should be corrected.

Response: Corrected

This study presents a valuable contribution to the field of sports science and injury prevention in professional soccer. However, refinements to the methodology, expanded discussion on limitations, and improved adherence to reporting guidelines are recommended to elevate the quality and impact of the manuscript

---

## [Editor Report · Decision Letter 2]

2 Dec 2024

Contractile and mechanical properties of hamstring muscles measured by the method of tensiomyography (TMG) in professional soccer players: A systematic review, meta-analysis and meta-regression

PONE-D-24-11496R2

Dear Dr. DANIEL FERNANDEZ BAEZA,

We’re pleased to inform you that your manuscript has been judged scientifically suitable for publication and will be formally accepted for publication once it meets all outstanding technical requirements.

Kind regards,

Holakoo Mohsenifar

Academic Editor

PLOS ONE
---

## [Editor Report · Acceptance letter]

PONE-D-24-11496R2

PLOS ONE

Dear Dr. Fernández-Baeza,

I'm pleased to inform you that your manuscript has been deemed suitable for publication in PLOS ONE. Congratulations! Your manuscript is now being handed over to our production team.

Kind regards,

on behalf of

Dr. Holakoo Mohsenifar

Academic Editor

PLOS ONE